# Control of Pest Grasshoppers in North America [note 1]

**DOI:** 10.3390/insects11090566

**Published:** 2020-08-24

**Authors:** Wahid H. Dakhel, Stefan T. Jaronski, Scott Schell

**Affiliations:** 1Department of Veterinary Pathobiology, College of Veterinary Medicine, University of Missouri, Columbia, MO 65211, USA; 2Department of Agriculture, USDA, Agriculture Research Service (ARS), Sidney, MT 59270, USA; thebugdoc01@gmail.com; 3Department of Ecosystem Science and Management, College of Agriculture and Natural Resources, Laramie, WY 82071, USA; sschell@uwyo.edu

**Keywords:** grasshopper, biological control, chemical control, bait formulation, entomopathogenic fungi

## Abstract

**Simple Summary:**

Grasshoppers (Orthoptera: Acrididae) population outbreaks occur frequently and consume damaging amounts of livestock forage and crops on millions of hectares of the western USA. The main method of controlling grasshopper outbreaks there consists of aerially applied spray with chemical insecticides. Although it is relatively cheap, fast, and efficient, broad spectrum insecticides can pose a threat serious risks to human health, and non-target organism populations which impacts the environment. As an alternative, the use biological control organisms more specific to pest grasshoppers is a less environmentally hazardous alternative to traditional, synthetic insecticides. This paper reviews the many different (viral, bacterial, fungal) insect pathogens and application methods that have been tested as alternatives to synthetic insecticide sprays to manage pest grasshopper populations.

**Abstract:**

Grasshoppers (Orthoptera: Acrididae) frequently inflict damage on millions of hectares of western rangelands and crops. The main method of controlling grasshopper outbreaks consists of covering their infestations with chemical insecticides. Although it is relatively cheap, fast, and efficient, chemical control bears serious risks to human health, non-target organisms, and the environment. To overcome this challenge, biological control is a less environmentally hazardous alternative to traditional, synthetic insecticides. This paper reviews strategies that could be used as effective ways to control such pests with a special focus on effective bait formulations that might provide a key model in developing biological control strategies for the grasshopper population.

## 1. Introduction

Grasshoppers (Orthoptera: Acrididae) are the dominant herbivores in grassland ecosystems worldwide [1]. In the western United States (US), grasslands comprise nearly 262 million hectares. Around 400 described species of grasshoppers inhabit the 17 contiguous western states [2], with over 100 species in Wyoming [3,4]. Most grasshopper species are either harmless or beneficial to us. Although grasshoppers can negatively affect other herbivores by outcompeting with them for forage, grasshoppers also increase rangeland productivity by stimulating plant growth and accelerating nutrient cycling [5]. Grasshoppers are also a food source for many rangeland predators [6]. However, from time to time, some species may reach extremely high densities and cause economic injury to rangeland forage and cultivated crops [4], to the extent that grasshoppers are the most important agricultural pests in rangeland habitats across the western United States (US) [7]. Currently, the primary control tools are chemical insecticides, especially carbaryl, malathion, and diflubenzuron [3]. These chemical insecticides have a potently serious effect on non-target insects, mainly pollinators, environmentally sensitive areas, and endangered/threatened species [5]. Biological control would be a particularly valuable tool for use on rangeland and natural habitats due to the minimal effect on humans and livestock health [3]. This review will describe control strategies that focus on chemical and biological control with an emphasis on the use of effective bait formulation for grasshopper control.

## 2. Rangeland Grasshoppers (Orthoptera: Acrididae) and Their Economic Importance

Approximately 15 of the 400 western US grasshopper species are considered major economic pests of either range or croplands [8]. Nationwide, four grasshoppers’ species are responsible for about 90% of all grasshopper crop damage [2,9], *Melanoplus sanguinipes*, *M*. *bivittatus*, *M*. *differentialis*, and *M*. *femurrubrum* (Figure 1, Figure 2, Figure 3 and Figure 4). Most grasshoppers are highly polyphagous and feed on a wide variety of plant species [10].

They can consume as much as their total body weight per day, but this amount varies with the species and developmental stage [11]. Annually, forage losses to grasshoppers in the western US are around 25%, which exceeds damage from all other rangeland arthropod families [3,9,12]. Historical records of severe grasshopper damage in North America are documented from the second half of the 19th century, when devastating Rocky Mountain locust (*Melanoplus spretus*) swarms decimated crops and rangelands from Central Canada to Texas [13]. From 1874 to 1877, *M*. *spretus* infestations were extremely expansive and severe over large areas of the Great Plains, which led to the institution of the United States Entomological Commission by Congress to study and control grasshopper plagues [14]. Later on, in the 1930s, grasshopper outbreaks covered millions of hectares of federally and privately-owned land in 17 western states [14]. In 1972, the implementation of publicly supported control programs on rangelands was authorized by the United States Department of Agriculture, Animal and Plant Health Inspection Service (USDA-APHIS) [15]. One of its functions was to prevent severe grasshopper rangeland damage [15]. Forage losses to grasshoppers could be overwhelming: for example, in 1985, grasshopper populations damaged about 22.2 hectares of western rangeland and inflicted economic losses estimated at over $393 million [16].

The economic importance of pest grasshoppers is not limited to forage damage. During outbreaks, vast infested territories were treated with a pesticide, which involves considerable monetary and environmental costs. During the 1986–1988 outbreaks, 4.1 L of insecticides were sprayed to cover 8.1 hectares of western rangeland with a program expenditure of $75 million [17]. Keeping grasshopper densities below economic thresholds is costly, and there is concern about the negative environmental effects of large-scale chemical control practices [18]. For this reason, in the late 1980s, the USDA-APHIS initiated a multi-million-dollar research project entitled the Grasshopper Integrated Pest Management project (GHIPM) [18].

The goal of GHIPM was to reduce the economic cost of grasshopper management and decrease the negative effects of large-scale insecticide treatments. GHIPM provided pest managers with better guidelines for determining economic thresholds for publicly supported control measures and estimating the environmental impact of grasshopper control [16,18]. Theoretically, the economic injury level (EIL) is the “level of pest population at which the damage from pests becomes equal to the cost of control”, as defined by Headley [19], which is expressed in grasshopper density per unit area and fluctuates based on benefits and costs. Historically, grasshopper management decisions were based on an action threshold density of ten or more grasshoppers per square yard, which has been used for more than 50 years [19]. Based on that threshold, densities at or above that level indicate that management action should be considered [4].

## 3. Strategies for Managing Grasshoppers

### 3.1. Chemical Control

For grasshopper population suppression, chemical insecticides are often the most efficient and effective [20,21]. The products and techniques used for grasshopper control have evolved since their first uses in the late 19th century. The application of poisoned baits was the predominant approach until the mid-20th century [21]. From the 1880s to the 1930s, pest managers used bait with toxicants such as copper (II), acetoarsenite, and sodium arsenite. Subsequently, bait producers began to use chlorinated hydrocarbons, such as aldrin, and later, organophosphates such as malathion and carbamates such as carbaryl came to be the agents of choice [22]. Applications of these and other broad-spectrum chemical insecticides were very efficient; however, they led to negative environmental impacts [22,23]. In attempts to minimize both the cost and adverse effects of insecticides on the environment, including non-target insects and birds, Lockwood and Schell developed the Reduced Agent and Area Treatments (RAATs) strategy in 1997 [24]. Thus, instead of blanketing the pest infestations with maximum label rates of the insecticides, the rates are decreased from the label maximum, and treated swaths are alternated with untreated swaths. The strategy was developed to use malathion (many generic brands), carbaryl (Sevin^®^ and generic brands), and most of all, diflubenzuron (Dimilin^®^ 2 L), which are EPA-registered chemicals intended for the treatment of grasshopper infestations on rangeland [20].

In 2003, the Dimilin RAATs method was applied in eastern Wyoming to protect 0.2 hectares from pest grasshopper species assemblages and has been shown to be very effective [17,24]. The RAATs method is best used with an insecticide that has sufficient residual effectiveness, allowing grasshoppers to move into the treated swaths from the untreated areas while it is still efficacious. As an Insect Growth Regulator (IGR) which interferes with immature insect molts, Dimilin is currently the product of choice in both federal and private RAATs grasshopper control programs in the western United States. RAATs reduce the cost of application as well as the amount of insecticide needed, thus making control products more economically and environmentally viable [17]. In 2010, Dimilin RAATs were successfully applied to almost 2.4 hectares of Wyoming rangelands to control a severe grasshopper outbreak, with an average cost of only $1.25 per protected hectare [25].

A short review including the three main insecticides currently used in grasshopper control in the western United States is presented below with the aim of highlighting their pros and cons.

#### 3.1.1. Carbaryl

Carbaryl (1-Naphthyl-n-methylcarbamate) is efficacious against grasshopper adults and nymphs. It kills grasshoppers that ingest it rather than those that come into physical contact with it [20]. It acts by suppressing the Central Nervous System (CNS) neurotransmitter acetylcholinesterase (AChE) [20], which typically provokes convulsions, paralysis, and eventually death of the treated insect. In 1956, Union Carbide created carbaryl and registered the treatment for numerous insects in forests, rangeland, pastures, for indoor plants and some domestic animals [26]. Carbaryl has moderate to acute oral toxicity to mammals: it’s LD50 (the amount which causes the death of 50% of a group of test animals) is 500 mg per kg of body weight for rats [26]. Although carbaryl has moderate toxicity to fish and limited toxicity to birds, it is exceedingly toxic to aquatic invertebrates and numerous arthropods, including beneficial species such as bees and other pollinators [25]. Carbaryl has been classified as a “possible human carcinogen” because of moderate acute oral toxicity to humans [25].

Carbaryl is relatively persistent in rangeland ecosystems; its residual action may continue for 3–10 days depending on the initial application rate [20]. It is most effective at temperatures ranging from 15.5 to 26.6 °C; at lower temperatures, the insecticide works very slowly [26]. After spraying carbaryl in the first two days, grasshopper mortality may range from 30 to 80% depending on environmental conditions. However, mortality may reach 90% under optimal application conditions [26].

Currently, carbaryl is used as a spray at 0.6 to 1.1 kg of active ingredient per hectare or, more commonly, as poisoned bait. It is the only chemical insecticide which is registered for use in bait formulation in the United States for control of many species of insects on rangelands, forests, pastures, and can be applied at 4.5359 kg (0.2268 kg of active ingredient) of 5% carbaryl bait per hectare in a blanket coverage or at 4.5359 kg (0.0907 kg of active ingredient) of 2% carbaryl bait per hectare in a RAATs application [26].

#### 3.1.2. Malathion

Malathion is the common name of the 0,0-dimethyl phosphorodithioate ester of diethyl mercaptosuccinate, developed by American Cyanamid in 1950. Because of its effectiveness, malathion is registered for use on many pests of rangeland, pastures, fruit, and vegetable crops, as well as for some medical and veterinary uses. The acute oral LD50 of malathion for rats is 1.375 mg per kg of body weight, which means that it ranges from being slightly to moderately toxic to mammals [20]. It is not very toxic to most bird species; however, it is extremely toxic to fish and aquatic invertebrates [20]. As was the case with carbaryl, malathion affects the function of CNS AChE [20]. Grasshoppers and beneficial arthropods such as bees and other pollinators are highly susceptible to malathion. As a non-selective insecticide, malathion can have negative effects on natural grasshopper enemies [27]. Malathion kills insects via two mechanisms, through direct contact or ingestion. The combined mechanisms result in high insect mortality [27]. Malathion is fast-acting, producing the majority of its control in the first two days post-application. Residual activity declines rapidly two to five days after treatment. As seen with carbaryl, malathion is also effective on warm days, withstanding temperatures greater than 26.6 °C. There are numerous formulations of malathion including Cythion^®^, Fyfanon^®^, and Malathion^®^, used for large-scale, USDA-APHIS-managed grasshopper treatment programs. The 0.2366 L rate per hectare is the standard conventional spray rate. Rates as low as 0.1182 L per hectare have worked in suitable weather conditions and in combination with the RAATs technique [27].

#### 3.1.3. Diflubenzuron

Diflubenzuron is an IGR and can be used for rangeland grasshopper as well as other arthropod pest management. Unlike broad-spectrum neurotoxic insecticides (e.g., carbaryl and malathion) that have been registered for use on rangeland grasshoppers, diflubenzuron interferes with the synthesis of chitin, which plays a critical role in the formation of the insect exoskeleton during molting, leading to insect death [28].

Diflubenzuron affects many pests in their immature stages, including mosquito larvae, moth caterpillars, and grasshopper nymphs. Adult insects that consume diflubenzuron can survive, but their eggs will often be less viable [28]. With an LD50 of 4.640 mg per kg of body weight for deer mice, diflubenzuron has extremely low toxicity to mammals [20]. It also has low toxicity to birds and fish [20]. However, diflubenzuron can have negative impacts on non-target invertebrate species in freshwater aquatic environments [29]. Diflubenzuron can be applied earlier than malathion and carbaryl because it has a four-week residual activity, so it is not necessary to wait until all grasshoppers have hatched before treating them [29]. This is advantageous, because if treated too early with malathion or carbaryl, grasshoppers may hatch after the insecticide’s residual activity has expired [29].

Diflubenzuron was registered for use on rangeland grasshoppers in 2001 and soon became the preferred pesticide by USDA-APHIS for grasshopper control programs [30]. The commercial Dimilin^®^ 2 L formulation of diflubenzuron is compatible with ultra-low volume (ULV) applications, which is the most efficient spray pesticide against grasshoppers [30]. An amount of 0.1 kg of the Dimilin^®^ 2 L (7.2 mg of active ingredient) per hectare is the most commonly used rate. The regular Dimilin 2 L application formula includes a minimum of 0.6 L of water and 0.3 L of crop or vegetable oil adjuvant per treated hectare. This makes it ULV-compatible with small volume, single-engine spray planes. Rates as low as 0.05 L of Dimilin^®^ 2 L (0.0054 kg of active ingredient) per hectare have been used successfully with the RAATs strategy [25]. As mentioned, in 2010, a large-scale Dimilin-RAATs grasshopper control program was successfully applied to almost 2.4 hectares of Wyoming rangelands [25].

### 3.2. Biological Control

The biological control of pests is an attractive alternative to chemical pesticides. Biological control is defined as a process of using organisms or microorganisms such as parasites, predators, or pathogens to suppress high pest population densities below the EIL [31]. The microbial biocontrol agents (MBCAs) such as pathogens, together with grasshopper parasitoids and predators, are numerous, and at times play an integral role in limiting grasshopper populations to low densities with little hazardous effects on humans and their environments. Throughout the world, some MBCAs are used with great advantage and success on insect pests [31]. A variety of approaches can be utilized when implementing biological control, such as conservation biocontrol, augmentation biocontrol, and classical biocontrol [32].

Conservation biological control is best done by managing the natural or agricultural ecosystem in a way that favors or at least preserves naturally occurring biological control organisms of the major pest species. For example, by not using herbicides on flowering, non-crop plants, pest managers allow the plants to provide a nectar source for syrphid fly (Diptera: Syrphidae) adults, the larvae of which are voracious predators of aphids [33].

In augmentation biological control, the biocontrol agents are collected where they are abundant or are reared artificially and then released on the pest. This can be divided into two approaches: inoculative and inundative [32]. An inoculative release relies on the agent to persist, naturally reproducing and distributing itself throughout the habitat of the pest to eventually reduce the pest’s population below the economic threshold. An inundative release is when a substantial number of the biocontrol agent is released to suppress the pest insect population in a short period of time, causing a nearly instantaneous reduction of the pest infestation [32].

Classical biological control refers to the suppression of an exotic insect pest infestation by introducing its natural enemies from the pest’s geographic origin [32]. The use of a parasitoid, such as the hymenopteran egg parasitoid *Scelio* spp., is one of the successful classical biological controls of grasshoppers [34]. For instance, when a *Scelio* spp. from Malaysia was introduced to Hawaii, it led to a reduction of grasshoppers to levels below the EIL [35]. A concern about using a foreign parasitoid to control US grasshoppers prevented the granting of permission to import *Scelio* spp. for grasshopper control [34]. However, there are examples of using exotic biocontrol agents against US grasshoppers. The fungal pathogen *Entomophaga grylli* Pathotype 3 (*E*. *praxibuli*) was imported from Australia and released in the western US as a potential grasshopper biological control agent. *E*. *praxibuli* infects melanopline and oedipodine grasshoppers. The agent may be of limited value because it later disappeared from its release sites [36]. Augmentation biological control that uses arthropod predators and parasitoids of grasshoppers is not a practical alternative due to the difficulties of rearing large numbers of arthropods in captivity, as well as the difficulties of releasing them onto immense areas of grasshopper outbreaks. The use of MBCAs (native or exotic but long established in North America) may be a useful approach and a viable alternative to the chemical pesticides currently available to control rangeland grasshoppers. Such MBCAs include viruses, bacteria, nematodes and fungi.

#### 3.2.1. Viruses

Viruses are one of the most actively investigated groups of insect pathogens [37]. Only three groups of viruses have been isolated from grasshoppers. Both entomopoxvirus (EPV) and crystalline array virus have been extracted from *Melanoplus* species in the US by Henry and Jutila [37]. The third virus is cytoplasmic polyhedrosis virus (CPV), first isolated from the grasshopper *Caledia captiva* in Australia [37]. Since the original detection in *Melanoplus sanguinipes*, EPVs have been identified from nine grasshopper species throughout the world. They are efficacious against grasshoppers and crickets as biological control agents [37]. All of these entomopathogenic viruses must be ingested by the susceptible host. In the insect gut, the inclusion bodies dissolve, liberating the virion that infects the gut epithelial cells and then replicates in those cells’ nuclei. The infection can be expanded into host hemocoel, resulting in host death in 5–7 days [38].

The baculovirus family includes nucleopolyhedrosis viruses (NPVs) and granuloviruses, (GVs). Baculoviruses possess a circular double-stranded genome (dsDNA) contained in inclusion bodies (nuclear or cytoplasmic aggregates). Hosts of baculoviruses span a wide range of Hymenoptera, Diptera, and some Crustacea, although Lepidoptera are their primary hosts [39]. In the US, NPVs were used by the US Forest Service and USDA-ARS as microbial agents against gypsy moths (*Lymantria dispar*) [20]. In the Netherlands, NPV is commercially available for insect pests of greenhouse-grown flowers [20]. However, no NPVs have been found that affect grasshoppers and efforts for grasshopper control with viruses have focused on EPVs.

Based on the structure of virions, the subfamily Entomopoxvirinae has been classified into three genera: *Entomopoxvirus* A, from Coleoptera; *Entomopoxvirus* B, which is restricted to Orthoptera and Lepidoptera; and *Entomopoxvirus* C, found in Diptera [40]. *Entomopoxvirus* B is able to infect 15 species of grasshoppers and locusts. The migratory grasshopper’s *Melanoplus sanguinipes* EPV has been extensively investigated and it infects two related species: *M*. *differentials* and *M*. *packardii* [41]. Horizontal transmission of EPVs to other individuals in a population may occur via feeding upon infected cadavers [37]. However, small field evaluations with an EPV indicate it is too expensive to implement on a larger scale [37].

#### 3.2.2. Bacteria

Entomopathogenic bacteria are a very diverse group of insect pathogens with more than 90 naturally occurring species [42]. Generally, bacterial pathogens infect their host *per os* during feeding. Bacterial pathogens can be isolated from dead insects, plants, and soil [42]. *Coccobacillus acridiorum* was the first bacterial pathogen used as a biological control agent of grasshoppers in Mexico early in the 20th century. Later, it was discovered that *C*. *acridiorum* also affects warm-blooded animals so its use was discontinued for grasshopper control [43]. The gram-negative bacterium, *Serratia marcescens*, is entomopathogenic bacterium. It leads to internal and external symptoms and signs of disease, including red diarrhea, and produces red pigments that present on the insect body (Figure 5).

The bacterium was originally extracted from the desert locust *Schistocerca gregaria* in Kenya and is now a well-established pathogen of laboratory grasshopper and locust colonies [44]. Perhaps the best known and most widely used entomopathogenic bacterium is *Bacillus thuringiensis* (*Bt*). *B*. *thuringiensis* is a gram-positive bacterium isolated from soil, plants, and the guts of various Lepidoptera and Coleoptera [45]. *B*. *thuringiensis* produces proteinaceous crystal toxins during sporulation known as δ-endotoxins. After the bacteria are ingested, these δ-endotoxins solubilize into active proteins within the highly alkaline midgut lumen [45]. Activated toxins bind to midgut epithelial cells through specific receptors and create perforations, allowing toxic material to pass through the membrane of midgut columnar cells. Ultimately, the host dies as a result of starvation or septicemia [45]. *B*. *thuringiensis* expresses two main types of protein toxins: crystal and cytoplasmic [46,47]. Currently, there are 73 known crystal toxin subtypes, highly specific to their targets, including Lepidoptera, Coleoptera, Diptera (mosquito larvae, black fly larvae, shore fly larvae etc.), and nematodes. Unfortunately, *Bt* is not lethal to grasshoppers due to the acidic environment of their midgut, which does not allow the crystal toxins to dissolve. However, recently, a new *Bt* endotoxin (Crystal 7A) that appears lethal against acridids has been discovered [46,47].

#### 3.2.3. Nematodes

Entomopathogenic nematodes are soft-bodied, non-segmented roundworms, some of which are parasites of insects. Entomopathogenic nematodes associated with grasshoppers are parasites that live inside the grasshopper’s body (endoparasites) [48]. Two nematodes, *Mermis nigrescens* and *Agamermis decaudata*, are in the family Mermithidae, and commonly parasitize grasshoppers [48]. Grasshoppers can be infected by *M*. *nigrescens* via their digestive tract through ingestion of the nematode eggs that have been laid on vegetation. However, *Agamermis decaudata* larvae enter a host grasshopper by simply coming into contact with and then penetrating its cuticle [41]. Once the larvae of entomopathogenic nematodes reach a grasshopper’s hemocoel, they consume its nutrients in order to complete their development. After that, they emerge from the grasshopper to complete their life cycle in the soil. When the adult nematodes exit from the grasshopper’s body, the grasshopper dies [48].

To date, more than 85 species of entomopathogenic nematodes have been described from two families, Heterorhabditidae and Steinernematidae [49]. Entomopathogenic nematodes *Steinernema* and *Heterorhabditis* have been widely used as MBCAs to manage several important insect pests [50]. These entomopathogenic nematodes harbored symbiotic bacteria, *Xenorhabdus* spp. and *Photorhabdus* spp., respectively, in the gut [51]. During infective stages, the juvenile larvae of these nematodes seek out their host and reach the hemocoel via the host’s natural openings like spiracles, the mouth, or the anus. Once in the hemocoel, the symbiotic bacteria will be released by the nematode and kill the host within 24 to 48 h [51]. Subsequently, infected cadavers provide shelter and food for the nematode’s growth, and then nematode migrates out of the cadaver and into soil [51]. Different nematode species have differing efficacy against grasshoppers. [52] Nicolas et al. (1995) found that nymphs of *M*. *sanguinipes* were more susceptible to *Steinernema carpocapsae* than to *S*. *scapterisci* under laboratory conditions. However, despite extensive research, currently, nematodes are not used for grasshopper biological control because they have long life cycles and require high moisture conditions.

#### 3.2.4. Microsporidia

Microsporidia are obligate, single-celled parasites that can only reproduce in other living cells. For many years, these microorganisms have been recognized as potential biological control agents for pest insects. Historically, microsporidia were considered Protozoans, but more recent molecular studies have reclassified microsporidia within the kingdom Fungi [53]. Currently, 186 genera of microsporidia comprise the most important group of (protozoan) pathogens of insects [53]. Based on classic categorization, microsporidia belong to a eukaryotic phylum; microsporidia are intracellular, spore-forming parasites of vertebrate and invertebrate hosts [54]. Microsporidia infections can be transmitted on to the next generation of host insects via the eggs (transovarial transmission) as well as by feeding (ingestion of the spores) [54]. The infection by microsporidia can be acute, resulting in death in several days, or chronic, causing a decrease in fecundity or even sterility [55,56]. Female grasshoppers infected by the microsporidian *Paranosema locustae* produce fewer eggs compared to healthy females [56]. In general, microsporidia provide effective grasshopper control at the home-owner level, but not at the range level.

*Paranosema* (*Nosema*) *locustae* is a MBCA developed for controlling grasshopper populations. It was originally isolated from the African Migratory locust *Locusta migratoria migratorioides* [57]. Beginning in the late 1960s, *P*. *locustae* was introduced to control rangeland grasshopper infestations in the US [57]. Subsequently, the use of *P*. *locustae* became registered for the long-term reduction of grasshoppers in the USA, but it is questionable regarding its efficiency and economics for large-scale grasshopper outbreaks [57]. It was also introduced for grasshopper control in Argentina [58].

Unlike other microsporidia, *P*. *locustae* has an unusually wide host range within Orthoptera [59]. In fact, natural or induced susceptibility to *P*. *locustae* has been recorded for as many as 102 orthopteran species [59]. Although *P*. *locustae* is advantageous in that it can specifically target pest species, it can also infect non-pest and rare grasshopper species [20]. As such, its large-scale use appears to be questionable in terms of a preservation of biodiversity standpoint.

*P*. *locustae* is typically formulated on wheat bran bait [60]. Bait consisting of only *P*. *locustae* causes a relatively small population reduction, no more than 12%. It caused a 58% reduction of grasshopper pest densities when formulated with carbaryl [61]. In a different study by Johnson and Henry [62], *P*. *locustae* combined with carbaryl was responsible for a 76% decrease in Canadian grasshopper populations.

Bait with *P*. *locustae* spores is readily ingested by grasshoppers. The microsporidian infects the alimentary tract and/or fat body [63]. Primarily, the infection of *P*. *locustae* spores begins in the grasshopper midgut, and then spreads to the fat body. Once these spores are ingested, they activate as soon as they reach the grasshopper midgut. Spores will germinate in the body to affect tissues and hemocytes [63]. The infection occurs through the sporoplasm, which is inoculated by the spores through a polar filament, or “extended tube” into the midgut epithelial cell. The spore contents are deposited inside the midgut epithelial cell, where the reproduction of microsporidia takes place [63]. The *P*. *locustae* infection develops slowly: grasshoppers start to show disease symptoms several weeks after treatment [64]. As mentioned before, host mortality caused by this microsporidian is usually relatively low, and the pathogen needs a long time to develop into a disease [3]. Although bait formulations using *P*. *locustae* against grasshoppers have been registered and commercialized in the US since the 1980s (NoLoc^®^, NoloBait^®^, Semaspore^®^ and other brands), their operational use was called into question [21]. The *P*. *locustae* products are frequently used to suppress pest grasshoppers in gardens; however, they failed from the standpoints of efficacy and economics when dealing with large-scale grasshopper outbreaks [64].

#### 3.2.5. Fungi

Historically, Ascomycete fungi such as *Beauveria*, *Metarhizium*, *Paecilomyces/Isaria*, and *Verticillium*/*Lecanicillium* were once classified in the Deuteromycetes (the Fungi Imperfecti, which do not possess the sexual structures) because the “perfect” (sexual) stages were not known. After molecular techniques were developed, the fungi were reclassified as those which possess “perfect” stages [65,66]. Ascomycota and Zygomycota are the most common entomopathogenic fungal groups [66]. As early as 1835, fungi were considered for use as a biological control of many pest species [65,66]. Insects are susceptible to more than 700 species of entomopathogenic fungi from approximately 90 genera. More than 170 mycopesticides were available as commercial products in 2007 worldwide. Currently, there are about ten commercial mycopesticide products registered by U.S. Environmental Protection Agency (EPA) [66,67].

Most insect pests are susceptible to fungal pathogens, of which *B*. *bassiana*, *B*. *brongniartii*, *Metarhizium anisopliae sensu lato*, *Isaria fumosorosea*, and *Entomophthorales* are the dominant species [66,67]. *B*. *bassiana* has a wider host range than *E*. *grylli*, which affects only grasshoppers. Approximately 37.2% of the commercial products sold in the early 2000s used *B*. *bassiana*, followed by *M*. *anisopliae* (36.4%), *Isaria fumosorosea* (5.8%) and *B*. *brongniartii* (4.1%) [66,67]. In contrast to bacteria, protozoa (including microsporidia in old classifications) and viruses, which need to be ingested by the target arthropod, entomopathogenic fungi penetrate the host cuticle upon contact [66]. Thus, fungi are more effective in infecting their hosts than most other entomopathogens [68]. Fungal infection begins when spores (conidia, blastospores) come into contact with the cuticle of susceptible hosts. There the spores encounter stimuli that elicit germination of an appressorium on the cuticle surface, which produces a germination peg that penetrates the cuticle using mechanical pressure and a combination of enzymes [66,68].

Once the fungus penetrates the epidermis and enters hemocoel, it proliferates by growing yeast-like hyphal bodies in Entomophorales fungi order or blastospores and mycelia in Ascomycetes, depleting nutrients in the hemolymph [66]. Eventually, death is caused either by fungus-produced toxins or digestive enzymes that destroy host tissues [66]. Under appropriate conditions (i.e., long periods of high humidity), the fungus grows out through the cuticle and produces conidia (Figure 6).

Several species of fungi are known to infect Orthoptera. *Beauveria bassiana* has been successfully field-tested for the control of rangeland grasshoppers using water and oil formulations, and it has been registered as a commercial product for that purpose in the US [69]. *Beauveria bassiana* is distributed worldwide and occurs naturally in the soil, causing “white muscardine disease” (Figure 7) [69].

In addition to *B*. *bassiana*, four additional species have been added to the genus later comprising *B*. *amorpha*, *B*. *caledonica*, *B*. *vermiconia*, and *B*. *velata* [66]. *B*. *vermiconia* and *B*. *amorpha* were extracted from soil and Coleoptera, while the *B*. *brongniartii* is particularly common in soil-inhabiting insects, especially Scarabaeidae [70]. *B*. *bassiana* is not commonly used against grasshoppers because of its low optimum growing temperature which allows grasshoppers to inactivate the fungus when they raise their body temperature via behavioral fever [66,70].

The genus *Lecanicillium* includes both *L*. *muscarium* and *L*. *longisporum* (both previously known as *Verticillium lecanii*) [66]. *Lecanicillium* has potential as an entomopathogenic fungus that is used in biological control of different pests [66]. These entomopathogenic fungi have been successfully developed and used as microbial control agents of various insects including Hemiptera, Homoptera, and spider mites, but they appear marginally effective against grasshoppers [67]. *L*. *longisporum* appears to severely infect soft scale insects and aphids and can reduce fungal plant pathogens (such as powdery mildew) on the leaves of cucumbers in a greenhouse [71].

Entomophthorales have been historically classified within Zygomycetes, but in the last decade have been placed in the subphylum Entomophthoromycotina [66]. The *Entomophaga grylli* (Fresenius) Batko pathotype or species complex is a major entomopathogenic fungus affecting acridids [66]. The commercial formulation of a bioinsecticide based on *E*. *grylli* is seriously compromised by the necessity to mass-produce the fungus in vivo, as it does not grow on artificial media. The *E*. *grylli* complex consists of three pathotypes: pathotypes 1 and 2, *E*. *macleodii* and *E*. *calopteni,* respectively, are pathogens of the acridid subfamilies Oedipodinae and Melanoplinae in North America. Pathotype 3, *E*. *praxibuli* infects the same subfamilies in laboratory studies and was introduced from Australia to possibly manage grasshopper pests in the US [36,66]. The *E*. *grylli* complex causes “summit disease”, which involves stimulating infected grasshoppers to climb to the top of a grass stem before killing them [36].

*Metarhizium* is another well-known genus of entomopathogenic fungi belonging to Phylum Ascomycota, Order Hypocreales, Family Clavicipitaceae. In 1879, the species *M*. *anisopliae* was described from scarabaeid beetles and originally defined by Metschnikoff as *Entomophthora anisopliae* [43]. Later, in 1883, Sorokin established it as the cause of the so-called “green muscardine fungus disease” due to the green color of its fungal spores. *Metarhizium* spp. fungi infect a broad range of insects including Homoptera, Acari, Hymenoptera, Lepidoptera, Diptera, Orthoptera and Coleoptera and their virulence depends on the host species [72].

Presently, approximately 10 to 15 common species comprise Metarhizium: *M. anisopliae*, *M. flavoviride*, *M. globosum*, *M. brunneum*, *M. majus*, *M. acridum*, *M. robertsii*, *M. guizhouense*, *M. pingshaense*, *M. frigidum* [72]. For more than 130 years, *M. anisopliae* has been used against many common insect pests [73]. Generally, *Metarhizium* spp. infect a wider range of insect species than *Beauveria* spp. However, there are certain species, such as *M. acridum*, which are highly specific to locusts and grasshoppers (Acrididae), and thus have a narrower host range compared to *Beauveria* spp. [72,73].

*M*. *acridum*, which has been isolated from locusts and grasshoppers, has been used for biological control of these pests throughout Africa, Australia, Madagascar, Canada, Brazil, and Mexico [72,73]. In Africa, *M*. *acridum* strain IMI330189 has been used for locust control under the name of Green Muscle^®^, while in Australia another strain, FI985, has been commercialized as Green Guard^®^ [72]. *M*. *acridum* sprays, applied in various formulations (oil), demonstrate high efficacy against grasshoppers [66,72]. In the field, Ultra-Low Volume (ULV) spraying with oil formulations is one of the most effective methods of fungal application to control grasshopper populations, although bait formulations have also been effective [74].

The *M*. *acridum* infection process is identical to other entomopathogenic Ascomycetes. The mode of infection includes adhesion, germination, differentiation, and penetration. Adhesion involves a spore attaching to the cuticle wall, where it germinates to produce an initial hyphal tube (appressorium) that penetrates directly through the exoskeleton and epidermis to reach the hemocoel, in which it develops as yeast-like blastospores. The infection process involves mechanical and enzymatic degradation of the cuticle. The host will be killed as a result of starvation, nutrient depletion, or body obstruction by the proliferation of the hyphal bodies [66,72]. Sporulation on insect cadavers happens in high humidity conditions [66], but more commonly the fungus dies with the insect, unable to emerge from the cadaver and produce conidia. Insecticidal metabolites, such as the cyclic polypeptide destruxins, are secreted by *Metarhizium* spp. to improve pathogenesis and successful reproduction [66,72].

Another entomopathogenic fungus, *M*. *brunneum* strain F52 (formerly *M*. *anisopliae sensu lato*) was recently registered in the US for controlling insects, including Coleoptera in horticulture and turf management and soft-bodied ticks [3]. It has been examined in both laboratory and field settings as a liquid formulation for the potential management of the Mormon cricket, *Anabrus simplex* (Orthoptera: Tettigoniidae). It caused higher mortality in the lab compared to the field, where neither spray nor bait applications were successful [75].

*M*. *anisopliae sensu lato* is applied to control locusts and grasshoppers, and negative side-effects on plants or birds have not been reported. Additionally, *M*. *anisopliae* had no negative impact on rabbits or frogs [73]. Apparently, *M*. *anisopliae* does not have mammalian toxicity, although one exception was reported by Mycotech Corporation in USA. They reported quick toxicosis and mortality in mice, which were inoculated by the intranasal-pulmonary route with the conidia from two different isolates of *M*. *anisopliae* collected from Madagascar [38]. Some mild adverse effects on non-target insects have occurred when *Metarhizium* was applied in the field, but non-target effects of the fungus in the field are less common than in the laboratory [47,72]. Furthermore, there have been a few cases of immunocompromised people infected with *Metarhizium* and *Beauveria*. It is recommended that people with fungal allergies and those who handle the fungal spores use adequate protection when working with entomopathogenic fungi [38,73]. The efficacy, germination, growth, and longevity of *Metarhizium* conidia can be affected by numerous environmental factors, such as temperature, humidity, and solar radiation. Generally, the entomopathogenic fungi can withstand a wide range of temperatures between 15 and 35 °C: although 25–29 °C is the optimal range for *M*. *anisopliae* germination and growth [66,72]. Relative humidity (RH) is an important environmental factor, affecting the efficacy and survival of the fungus. In particular, high relative humidity is usually necessary for *Metarhizium* conidia to germinate. [76]. Athanassiou et al. (2017) demonstrated that relative humidity for germination is limited to 92.3–100% [76]. Yet it is noted in the field that an oil-based formula used against Desert locusts has been recorded with high infection rates with low humidity (20–30% RH) [77]. Besides temperature and humidity, solar radiation acts in *Metarhizium* spore survival in the field [78]. Solar radiation has adverse effects on spore longevity and causes inactivation of *Metarhizium* conidia. Likewise, when conidia are protected under microhabitats such as a dense crop canopy, the shade encourages the survival of *Metarhizium*, and the fungus persists [73].

### 3.3. Bait Formulation

The keys for mycoinsecticide success lie in three main areas: formulation development, application, and understanding the biology of the host-pathogen relationship in the field. Formulation products have been a goal of grasshopper control for many years, and they come in different forms, such as dust, wettable powders, granules, and baits, liquid formulations formed from biomass suspensions in water or oils, or a mixture of solids and liquids in emulsions [66,67,72]. Dust and baits are regularly used for small-scale applications, and sprays are more suitable for large-scale operations [79]. Latchininsky et al. [21], 2006 mention that baits are safer than sprays for non-target organisms and for the applicator, but on a larger scale sprays are more cost-efficient. Bait formulations are cheaper than sprays for small-scale operations, and baits have a high potential because they may be improved in the future with added attractants and chemicals that protect the spores from UV exposure [21].

Baits are likely to be specific to the target organism, or at least be more selective than liquid and dust treatments [3,72]. Poisonous baits were the first efforts to control North American grasshopper outbreaks, and since the 1870s, testing has been conducted on these [80]. For over forty years, a common strategy for controlling grasshoppers in the western US has involved the use of carbaryl bait on wheat bran [81]. In the 1940s, an experiment was conducted in Russia on Siberian grasshoppers involving bait consisting of a wheat bran carrier with a fungal pathogen (*Entomophaga grylli*) that resulted in a reduction in grasshopper egg production by 72–92% [81]. Currently, in the US, treatments used to control grasshoppers by USDA-APHIS depend on spray and bait formulations, with carbaryl or spray formulation using diflubenzuron and, less commonly, malathion [20]. Due to the lack of using entomopathogenic baits with *M*. *anisopliae* and *B*. *bassiana*, more attention should be given to promoting fungal baits, bearing in mind their environmental advantages over sprays for the control of grasshopper pests.

## 4. Conclusions

This literature review provides an overview of rangeland pest grasshoppers, control of their populations, and their economic importance. Grasshoppers are serious pests of agriculture in western North America, and critical control is necessary for *food* security. Presently, the primary control tools are chemical insecticides, specifically diflubenzuron, carbaryl, and malathion. Pesticide control can be effective, quick-acting, and adaptable to all agricultural *conditions*. However, Federal grasshopper control programs are not authorized to use this pending issuance of the Environmental Impact Statements, which are not being pursued at this time. The existing broad-spectrum chemicals have use limitations where there are endangered or threatened non-target animals and plants. There is a pressing need for environmentally friendly yet efficacious control measures to manage grasshopper populations in these natural habitats. Therefore, using microbial pesticides as an alternative to the registered chemical pesticides is a relatively environmentally benign treatment tool for grasshopper hotspots. The biopesticide would be a useful tool for agricultural and environmental management systems.

## Figures and Tables

**Figure 1 insects-11-00566-f001:**
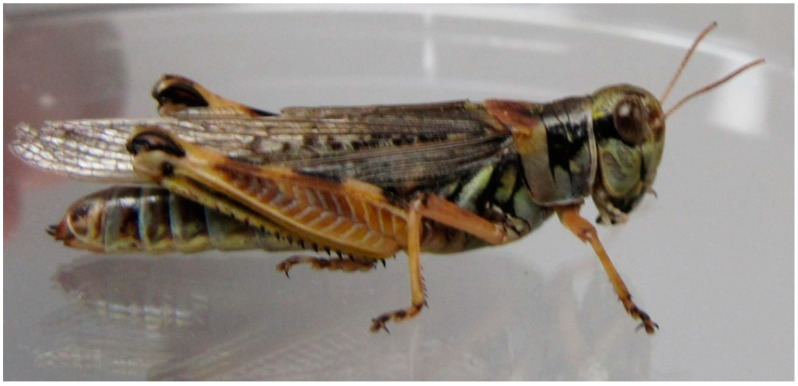
Female *Melanoplus sanguinipes* grasshopper. (Wahid Dakhel, 2014). University of Wyoming, Laramie, WY, USA.

**Figure 2 insects-11-00566-f002:**
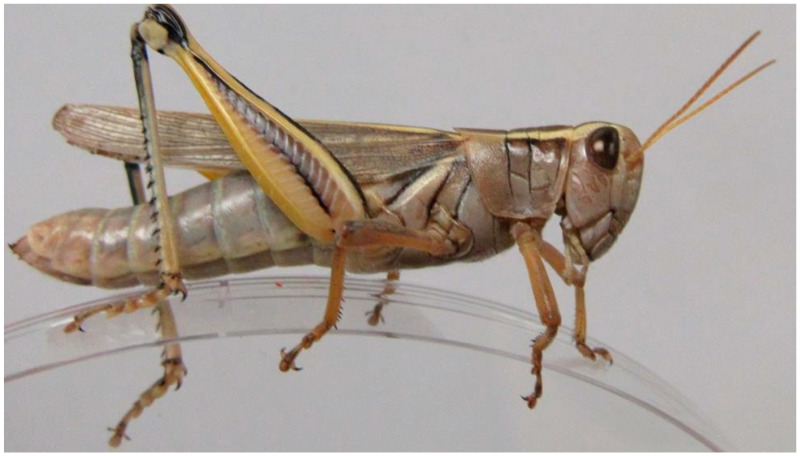
Female *Melanoplus bivittatus*. (Wahid Dakhel, 2014). University of Wyoming, Laramie, WY, USA.

**Figure 3 insects-11-00566-f003:**
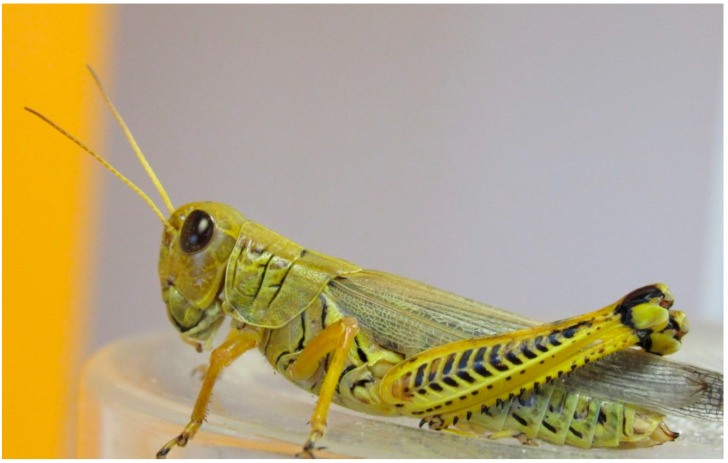
Female *Melanoplus differentialis*. (Wahid Dakhel, 2014). University of Wyoming, Laramie, WY, USA.

**Figure 4 insects-11-00566-f004:**
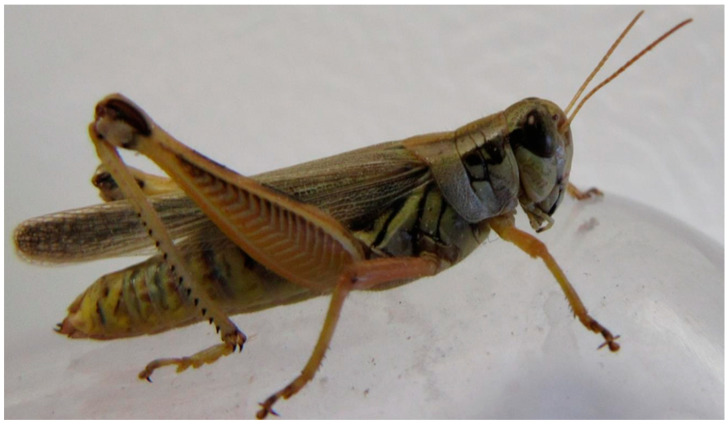
Female of *Melanoplus femurrubrum*. (Wahid Dakhel, 2014). University of Wyoming, Laramie, WY, USA.

**Figure 5 insects-11-00566-f005:**
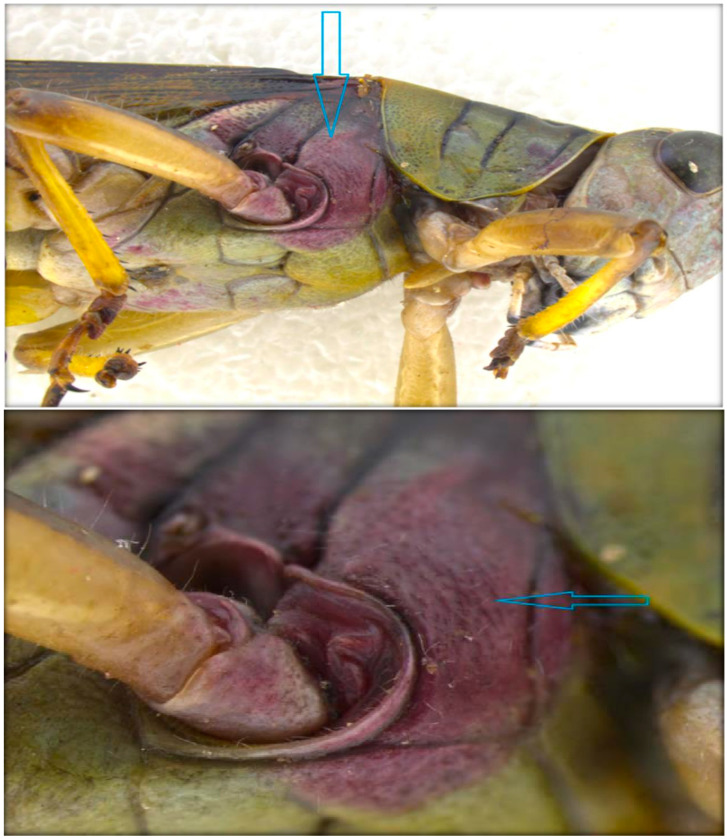
Adult female of *M. bivittatus* infected by *S. marcescens*. Note the distinct red tint on the thorax. (Wahid Dakhel, 2013).

**Figure 6 insects-11-00566-f006:**
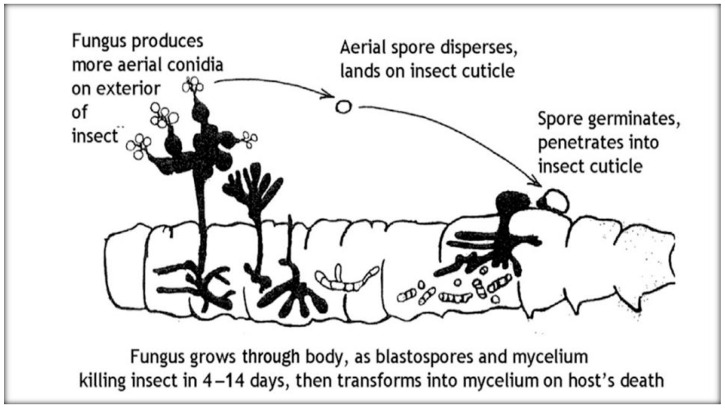
Schematic of the life cycle of entomopathogenic fungi [66].

**Figure 7 insects-11-00566-f007:**
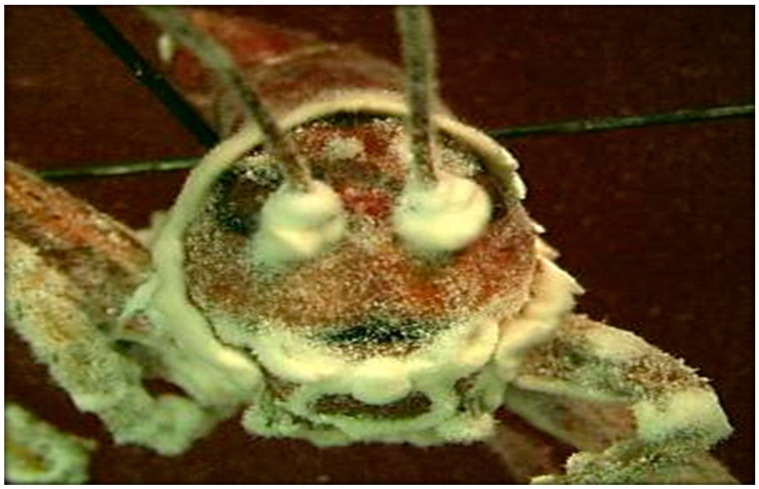
Beauveria bassiana emerging from the intersegmental of its host, a Mormon cricket (Anabrus simplex Haldeman), and conidiating on the surface of the cuticle [66].

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
