# Peer review of "Control of Pest Grasshoppers in North America"

_insects, 2020, doi:10.3390/insects11090566_

Round 1

Reviewer 1 Report

This paper review is well documented and analyzed for grasshopper control in North America, but there are some questions and minor suggestions for edition.

Author Response

Dear Editors,

Thank you very much for the review of our paper entitled: “CONTROL OF PEST GRASSHOPPERS IN NORTH AMERICA”. We sincerely appreciate all valuable comments and suggestions, which helped us to improve the quality of the article. Our responses to the Reviewers’ comment are described below in a point-to-point manner. An appropriated change, suggested by the Reviewers, has been introduced to the paper (highlighted within the document).

Best regards,

Reviewer 2 Report

The manuscript “Control of pest grasshoppers in North America” is a literary review of the technics and their efficiency to control pest grasshoppers. A focus is made on the US cases, especially in terms of authorized technics and products. Details are given related to chemical control and various types of biological control approaches.

I have some general and ‘line-by-line’ comments which, I hope, will help the authors to improve their manuscript.

General comments:

- An abstract is needed. Please see the Instructions for authors here: https://www.mdpi.com/journal/insects/instructions#preparation

- The Introduction must be deeply revised. The Introduction should present the context and justify why such a review is needed. It should also present the structure and objectives of the review. Please, also find instructions here: https://www.mdpi.com/journal/insects/instructions#preparation

- Some paragraphs are very general and do not focus on grasshoppers. I suggest that they must be deleted. References linking to more general knowledge can be provided so that readers unfamiliar with some technics will be able to read further. But the review here should keep focussing on effects on grasshoppers. Examples:

L. 177-L. 197: These general information about biological control are not needed here; readers will find this elsewhere. More importantly, there are no information about grasshoppers here.

L. 262-272: This paragraph is not needed if Bt is not an important tool against grasshoppers.

L. 344-370: If the biological processes and control technics described here are not specific to grasshoppers, I think that these paragraphs are not needed.

L. 387-394: Same comment here.

- Reference 20 is used 19 times in the paper, especially in the paragraph “Chemical control”. It is of concerns, because it looks as if the paper summarizes large part of the reference 20, which an original article should not do.

- Please revise all the numberings of the titles and subtitles.

Line-by-line comments:

L. 41: Figure 4 is missing.

L. 52: Please correct the sentence. There is a problem with the “in 17” here.

L. 59: I suggest to use “were” instead of “are”. Indeed, it was a choice to spread pesticides, and alternative tools could be used, as presented in the review.

L. 63: A reference is needed after “control practices”.

L. 78-79: Does the reference 21 really compare all the potential available control technics to support such an affirmation?

L. 116: A reference is needed after “other pollinators”.

L. 117: Please indicate what does “t0” refer to.

L. 121: Please provide temperature in °C as well.

L. 125: Please provide “fluid oz” in kg of active ingredients as well. Same comment at L. 168, L. 170.

L. 127: I think that “pounds” is not needed here.

L. 140: It could be interesting to have examples of grasshopper natural enemies here.

L. 144: “greater than” instead of “greaterthan” here. Also, please provide temperature in °C as well.

L. 174: Please clarify what does “successfully” indicate here.

L. 210-214: This is a personal opinion which I think should not be provided in such a review. Opinion/Forum papers are places for such personal comments.

L. 219: A reference would be needed after “insect pathogens”.

L. 221: Please adapt the style of the reference here. Same comment for Fig. 6 and 7; L. 380; 460; 474

L. 233: Please provide the scientific name (Latin name) of gypsy moths.

L. 246: I think that the first sentence here is not needed.

L. 289: Please rephrase this sentence: I don’t understand well what does “Infestive stages” mean in this sentence.

L. 296-297: It is a little strange to end the paragraph with such a sentence. Maybe, this sentence should come at the beginning of the paragraph, to explain that: despite nematodes are not (yet) used as biocontrol tools against grasshoppers, extensive research showed that…

L. 310-311: This sentence repeats the last one of this subpart.

L. 412-418: Please provide the scientific names in italic.

Figure 6 and 7: They seem to be taken from another paper published by the same authors. Please verify the authorization for reproduction.

Author Response

Dear Editors,

Thank you very much for the review of our paper entitled: “CONTROL OF PEST GRASSHOPPERS IN NORTH AMERICA”. We sincerely appreciate all valuable comments and suggestions, which helped us to improve the quality of the article. Our responses to the Reviewers’ comment are described below in a point-to-point manner. An appropriated change, suggested by the Reviewers, has been introduced to the paper (highlighted within the document).

Best regards,

Comments and Suggestions for Authors

The manuscript “Control of pest grasshoppers in North America” is a literary review of the technics and their efficiency to control pest grasshoppers. A focus is made on the US cases, especially in terms of authorized technics and products. Details are given related to chemical control and various types of biological control approaches.

I have some general and ‘line-by-line’ comments which, I hope, will help the authors to improve their manuscript.

General comments:

- An abstract is needed. Please see the Instructions for authors here: https://www.mdpi.com/journal/insects/instructions#preparation

- Abstract has been added to literary review

- The Introduction must be deeply revised. The Introduction should present the context and justify why such a review is needed. It should also present the structure and objectives of the review. Please, also find instructions here: https://www.mdpi.com/journal/insects/instructions#preparation

  • Sentences were added to the introduction

- Some paragraphs are very general and do not focus on grasshoppers. I suggest that they must be deleted. References linking to more general knowledge can be provided so that readers unfamiliar with some technics will be able to read further. But the review here should keep focussing on effects on grasshoppers. Examples:

  1. 177-L. 197: These general information about biological control are not needed here; readers will find this elsewhere. More importantly, there is no information about grasshoppers here.

- Some readers don’t have any idea about what is biological control and how it’s important, and most of them know the chemicals. We explain what is the biological control and how is different from chemical and at the end the talking specifically about grasshoppers

  1. 262-272: This paragraph is not needed if Bt is not an important tool against grasshoppers.

- We gave an idea That  Bt is not lethal to grasshoppers due to the acidic, so before that, we explain how lethal the bacteria is to reach to this point.

  1. 344-370: If the biological processes and control technics described here are not specific to grasshoppers, I think that these paragraphs are not needed.

- Entomopathogenic fungi consider being very important MOs to control grasshoppers and most of the researches have been recently focusing on the fungal pathogen, We gave a simple introduction and late explain the control technics for example bait formulation

  1. 387-394: Same comment here.

- Same comment

- Reference 20 is used 19 times in the paper, especially in the paragraph “Chemical control”. It is of concerns, because it looks as if the paper summarizes large part of the reference 20, which an original article should not do.

- Its right, the issue has been modified by adding more references

- Please revise all the numberings of the titles and subtitles.

- All the number has been adjusted

Line-by-line comments:

  1. 41: Figure 4 is missing.

- Figure 4 was added

  1. 52: Please correct the sentence. There is a problem with the “in 17” here.
  • The sentence was modified to “Later on, in the 1930s, grasshopper outbreaks covered millions of acres of federally and privately-owned land in 17 western states”

  1. 59: I suggest to use “were” instead of “are”. Indeed, it was a choice to spread pesticides, and alternative tools could be used, as presented in the review.

- Modification was added

  1. 63: A reference is needed after “control practices”.

 - A reference # [18] was added after “control practice”

  1. 78-79: Does the reference 21 really compare all the potential available control technics to support such an affirmation?

- In general not specifically as an idea

  1. 116: A reference is needed after “other pollinators”.

 - A reference # [25] was added after “other pollinators”.

  1. 117: Please indicate what does “t0” refer to.

 -t0 accidently happened as printing error on keyboard, and it has been removed

  1. 121: Please provide temperature in °C as well.

- Modification was added to all temperature in °C instead of °F “from 15.5 to 26.6 °C”

  1. 125: Please provide “fluid oz” in kg of active ingredients as well. Same comment at L. 168, L. 170.

 - All “fluid oz” was modified in Kg

  1. 127: I think that “pounds” is not needed here.

- “pounds” was removed

  1. 140: It could be interesting to have examples of grasshopper natural enemies here.

 - Natural enemies’ example has already added as a part of biological control instead of chemical control

natural enemies; natural enemies from the pest’s geographic origin. Use of a parasitoid, such as the hymenopteran egg parasitoid Scelio spp.,.

  1. 144: “greater than” instead of “greaterthan” here. Also, please provide a temperature in °C as well.

 - The word “greaterthan” was adjusted to “greater than”

  1. 174: Please clarify what does “successfully” indicates here.

 - Meaning the rates of Dimilin were very effective with the RAATs method.

  1. 210-214: This is a personal opinion which I think should not be provided in such a review. Opinion/Forum papers are places for such personal comments.

 - The personal opinion was removed

  1. 219: A reference would be needed after “insect pathogens”.

- A reference [37] was cited after “insect pathogens”.

  1. 221: Please adapt the style of the reference here. Same comment for Fig. 6 and 7; L. 380; 460; 474

 - A reference [66] was cited in figures 6 and 7

  1. 233: Please provide the scientific name (Latin name) of gypsy moths.

 - (Lymantria dispar) Latin name of gypsy moths was added

  1. 246: I think that the first sentence here is not needed.

 - The first sentence “Bacteria are prokaryotic, unicellular microorganisms “was removed prokaryotic, unicellular microorganisms.

  1. 289: Please rephrase this sentence: I don’t understand well what does “Infestive stages” mean in this sentence.

- The word “Infestive stages” was adjusted to “Infective stage”. Juvenile larvae are known as infective stage

  1. 296-297: It is a little strange to end the paragraph with such a sentence. Maybe, this sentence should come at the beginning of the paragraph, to explain that: despite nematodes are not (yet) used as biocontrol tools against grasshoppers, extensive research showed that…

- The end of the paragraph has been adjusted as However, despite extensive research, currently, nematodes are not used for grasshopper biological control because they have a long life cycle and required high moisture conditions

  1. 310-311: This sentence repeats the last one of this subpart.

 - Sorry, I could not find the repeats

  1. 412-418: Please provide the scientific names in italic.
  • Italics names of fungi were added

Figures 6 and 7: They seem to be taken from another paper published by the same authors. Please verify the authorization for reproduction.

  • These figures belong to Dr. Jaronski who is the second author, Mass Production of Entomopathogenic Fungi chapter 11, Jaronski, 2013

Reviewer 3 Report

This manuscript reviews grasshoppers control by chemical and biological control methods. It is very comprehensive, well written and clear. Some parts on principles of biological control, taxonomy, description and mode of action in individual group of biocontrol agents might look too general or as repetition of textbook and some details might not be necessary when not directly related by grasshoppers but on the other hand, particularly for readers not specialised in the field, it gives nice and understandable overview including history of control methods. I consider this submission very relevant for Insects journal and I am convinced that it will be frequently cited in future research papers. I have only few minor suggestions below.

Specific comments

Abstract and Keywords sections are missing (perhaps it is OK for review type of submission for Insects).

Numbering of sections is wrong and needs to be corrected (e.g. at line 28 „Rangeland grashoppers ...“ should have number 2., at line 149 „Diflubenzuron“ the number should be 3.1.3 and at line 491 „CONCLUSIONS“ should be 4. etc.).

It would be nice to mention purpose of/summarize the review content at the end of Introduction in 1-2 sentences.

Figure 3 requires editing to remove date stamp.

Figure 4 is missing

I suggest to combine photos in figures 1-4 into a single figure panel.

Line 52 some text seems to be missing in the middle of line

Line 107 - please check spelling 1-Naphthyl-N-methylcarbamate

Line 117 type “moderate t0” > “moderate to”

Line 119 the end should read “my continue for …”

Lines 121, 125 and onwards: please check the units, according to the journal guidelines “SI Units (International System of Units) should be used. Imperial, US customary and other units should be converted to SI units whenever possible.”

Line 127 0.23 kg pounds > 0.23 kg

Line 221 reference format (Street et al.) needs to be corrected

Line 288 These[space]entomopathogenic

Line 294 Reference format needs to be corrected

Line 305 should read parasites of vertebrate …

Lines 354-356 I suggest to write full species names when first mentioned, ie. Beauveria bassiana etc.

Line 368 main toxins and secondary metabolites might be mentioned along with references

Figure 6 was already published, the general schema of EPF cycle is well known and because no figure of e.g. Bt mode of action in insect gut is presented I would suggest to either delete it or move to supplemetary on-line material. It wold be worth to present photos of other EPF species causing mycosis on grasshoppers, e.g. Metarhizium in addition to Fig. 7.

Line 410 in addition Metarhizium infects also Acari

Lines 412-417 species names should be in italics

Paragraph between lines 427 and 433 might be shortened and the text describing infection process general to EPF can be moved to section starting at line 362 to avoid repetition.

Line 457 typo: although

Line 474 probably mistake in reference format

In section on EPF it might be also notice if there is any literature on Isaria infecting Orthoptera even it is not used for grasshoppers control, e.g. I found the following paper: A new species of Isaria isolated from an infected locust By: Liang, Zongqi; He, Xueyou; Han, Yanfeng; et al. MYCOTAXON Volume: ‏ 105 Pages: ‏ 29-36 Published: ‏ JUL-SEP 2008.

Author Response

Dear Editors,

Thank you very much for the review of our paper entitled: “CONTROL OF PEST GRASSHOPPERS IN NORTH AMERICA”. We sincerely appreciate all valuable comments and suggestions, which helped us to improve the quality of the article. Our responses to the Reviewers’ comment are described below in a point-to-point manner. An appropriated change, suggested by the Reviewers, has been introduced to the paper (highlighted within the document).

Best regards,

Abstract and Keywords sections are missing (perhaps it is OK for review type of submission for Insects).

  • Abstract and Keywords were added to the paper

Numbering of sections is wrong and needs to be corrected (e.g. at line 28 „Rangeland grasshoppers ...“ should have number 2., at line 149 „Diflubenzuron“ the number should be 3.1.3 and at line 491 „CONCLUSIONS“ should be 4. etc.).

  • All Numbering of sections were corrected.

It would be nice to mention purpose of/summarize the review content at the end of Introduction in 1-2 sentences.

  • The sentence at the end of the introduction were added

Figure 3 requires editing to remove the date stamp.

  • A new picture was added instead of the one with a date

Figure 4 is missing

- Figure 4 was added

I suggest to combine photos in figures 1-4 into a single figure panel.

  • If I understand regarding the size, it was adjusted

Line 52 some text seems to be missing in the middle of the line

  • The sentence was modified to “Later on, in the 1930s, grasshopper outbreaks covered millions of acres of federally and privately-owned land in 17 western states”

Line 107 - please check spelling 1-Naphthyl-N-methylcarbamate

  • Spelling check “(1- Naphthyl-N-Methylcarbamate)”

Line 117 type “moderate t0” > “moderate to”

-t0 accidentally happened as printing error on the keyboard, and it has been removed

Line 119 the end should read “my continue for …”

  • This was adjusted to may continue for.

Lines 121, 125 and onwards: please check the units, according to the journal guidelines “SI Units (International System of Units) should be used. Imperial, US customary and other units should be converted to SI units whenever possible.”

Line 127 0.23 kg pounds > 0.23 kg

  • The modification was added to all Units
  • Line 221 reference format (Street et al.) needs to be corrected
  • It has been cited as the number

Line 288 These [space]entomopathogenic

  • Space was removed

Line 294 Reference format needs to be corrected

  • At this point I have to mention the name, I cannot say 52 found that. OR I can write it has been that ….at the end put the number of references. Or a previous study found that….. What do you think?

Line 305 should read parasites of vertebrate …

  • Modification was added

Lines 354-356 I suggest writing full species names when first mentioned, ie. Beauveria bassiana etc.

  • In this case, I mentioned Names as general because there are many different kinds of Beauveria

Line 368 main toxins and secondary metabolites might be mentioned along with references

  • References # 66 was added

Figure 6 was already published, the general schema of EPF cycle is well known and because no figure of e.g. Bt mode of action in insect gut is presented I would suggest to either delete it or move to supplementary on-line material. It would be worth presenting photos of other EPF species causing mycosis on grasshoppers, e.g. Metarhizium in addition to Fig. 7.

  • This figure belongs to Dr. Jaronski who is the second author, Mass Production of Entomopathogenic Fungi chapter 11, Jaronski, 2013

Line 410 in addition Metarhizium infects also Acari

  • It was about Metarhizium spp. In general not acridum

Lines 412-417 species names should be in italics

  • Italics names of fungi were added

Paragraph between lines 427 and 433 might be shortened and the text describing infection process general to EPF can be moved to section starting at line 362 to avoid repetition.

  • First at line 362 there was comparing between mode of action” main rout of infection” of fungi with other organisms such as bacteria, Nematodes, and virus, while the paragraph at line 427-433, there was specifically description for the mode of action of fungi

Line 457 typo: although

  • Modification was added

Line 474 probably mistake in reference format

  • Reference format was adjusted

In section on EPF it might be also notice if there is any literature on Isaria infecting Orthoptera even it is not used for grasshoppers control, e.g. I found the following paper: A new species of Isaria isolated from an infected locust By: Liang, Zongqi; He, Xueyou; Han, Yanfeng; et al. MYCOTAXON Volume: ‏ 105 Pages: ‏ 29-36 Published: ‏ JUL-SEP 2008.                

  • Sorry, I did not do this because of the deadline for the resubmission.
